# Actions of Novel Angiotensin Receptor Blocking Drugs, Bisartans, Relevant for COVID-19 Therapy: Biased Agonism at Angiotensin Receptors and the Beneficial Effects of Neprilysin in the Renin Angiotensin System

**DOI:** 10.3390/molecules27154854

**Published:** 2022-07-29

**Authors:** Graham J. Moore, Harry Ridgway, Konstantinos Kelaidonis, Christos T. Chasapis, Irene Ligielli, Thomas Mavromoustakos, Joanna Bojarska, John M. Matsoukas

**Affiliations:** 1Department of Physiology and Pharmacology, Cumming School of Medicine, University of Calgary, Calgary, AB T2N 4N1, Canada; 2Pepmetics Inc., 772 Murphy Place, Victoria, BC V8Y 3H4, Canada; 3Institute for Sustainable Industries and Liveable Cities, Victoria University, Melbourne, VIC 8001, Australia; ridgway@vtc.net; 4AquaMem Consultants, Rodeo, New Mexico, NM 88056, USA; 5NewDrug PC, Patras Science Park, 26504 Patras, Greece; k.kelaidonis@gmail.com; 6NMR Facility, Instrumental Analysis Laboratory, School of Natural Sciences, University of Patras, 26504 Patras, Greece; cchasapis@upatras.gr; 7Institute of Chemical Engineering Sciences, Foundation for Research and Technology, Hellas (FORTH/ICE-HT), 26504 Patras, Greece; 8Department of Chemistry, National and Kapodistrian University of Athens, 15784 Athens, Greece; eir.ligielli@gmail.com (I.L.); tmavrom@chem.uoa.gr (T.M.); 9Institute of General and Ecological Chemistry, Faculty of Chemistry, Lodz University of Technology, Zeromskiego 116, 90-924 Lodz, Poland; joanna.bojarska@p.lodz.pl; 10Institute for Health and Sport, Victoria University, Melbourne, VIC 3030, Australia

**Keywords:** COVID-19, angiotensin II, angiotensin II receptors, biased agonism, molecular dynamics (MD), angiotensin II receptor blockers (ARBS), sartans, bisartans, ACE2, furin, 3CLpro, neprilysin, charge relay system (CRS), ARB/NEP docking

## Abstract

Angiotensin receptor blockers (ARBs) used in the treatment of hypertension and potentially in SARS-CoV-2 infection exhibit inverse agonist effects at angiotensin AR1 receptors, suggesting the receptor may have evolved to accommodate naturally occurring angiotensin ‘antipeptides’. Screening of the human genome has identified a peptide (EGVYVHPV) encoded by mRNA, complementary to that encoding ANG II itself, which is an inverse agonist. Thus, opposite strands of DNA encode peptides with opposite effects at AR1 receptors. Agonism and inverse agonism at AR1 receptors can be explained by a receptor ‘switching’ between an activated state invoking receptor dimerization/G protein coupling and an inverse agonist state mediated by an alternative/second messenger that is slow to reverse. Both receptor states appear to be driven by the formation of the ANG II charge-relay system involving TyrOH-His/imidazole-Carboxylate (analogous to serine proteases). In this system, tyrosinate species formed are essential for activating AT1 and AT2 receptors. ANGII is also known to bind to the zinc-coordinated metalloprotease angiotensin converting enzyme 2 (ACE2) used by the COVID-19 virus to enter cells. Here we report in silico results demonstrating the binding of a new class of anionic biphenyl-tetrazole sartans (‘Bisartans’) to the active site zinc atom of the endopeptidase Neprilysin (NEP) involved in regulating hypertension, by modulating humoral levels of beneficial vasoactive peptides in the RAS such as vasodilator angiotensin (1–7). In vivo and modeling evidence further suggest Bisartans can inhibit ANG II-induced pulmonary edema and may be useful in combatting SARS-CoV-2 infection by inhibiting ACE2-mediated viral entry to cells.

## 1. Introduction

Angiotensin II is a peptide hormone that mediates the main actions of the renin–angiotensin system (RAS), playing a crucial role in the control of blood pressure and fluid balance. Next, two forms of angiotensin were recognized, namely inactive decapeptide (D-R-V-Y-I-H-P-F-H-L) angiotensin I (ANG I) and powerful octapeptide (DRVYIHPF) angiotensin II (ANG II), the most potent pressor substance known. The vasoconstrictor ANGII peptide was formed through the rapid cleavage of inactive decapeptide ANG I by angiotensin-converting the enzyme (ACE1), dipeptidyl carboxypeptidase, or chymases [1,2,3], whereas ACE2 produces the vasodilator heptapeptide ANG (1–7) from ANGII. In addition, ANG II metabolites, such as ANG III, ANG IV, and ANG (1–7) should not be overlooked [4]. ANG II can bind to the ANG II receptors type 1 and 2 (AT1 and AT2) belonging to the G-protein-coupled receptors. However, most physiological influence is mediated by AT1. More specifically, ANG II acts at AT1 receptors in smooth muscle and other tissues, causing contraction and blood pressure elevation. Lowering ANG II levels with RAS inhibitors, such as ACE1 inhibitors and ANG II receptor blockers (ARB), by direct inhibition with AT1 receptor blockers, provides effective treatment of hypertension. ARBs demonstrate insurmountable blocking effects at AT1 receptors, which are like the effects seen with certain synthetic analogues of ANG II in which the C-terminal Phe is replaced by an aliphatic amino acid such as Ile. 

Notably, ANG II is a multifunctional bioactive short peptide. Its regulatory actions are related to not only vasoconstriction (constriction of the blood vessels), increased renal tubular absorption of sodium, and stimulation of adrenal aldosterone production but also to activation of the sympathetic nervous system or the pathogenesis of cardiovascular diseases [2]. ANG II, by AT1, causes cell growth and cellular phenotypic changes, regulates the gene expression of bio-substances, and activates multiple intracellular signaling cascades in cardiac myocytes and fibroblasts and endothelial and smooth muscle cells [3]. Interestingly, an ANG II vaccine, under clinical studies, can be a novel effective therapy against heart failure and hypertension [5,6]. The broad role of angiotensin II as a potent modulator of the immune system has been demonstrated in inflammation, immunity, rheumatoid arthritis, and multiple sclerosis [7,8]. Pioneer research by Steinman and collaborators has shown that blocking an angiotensin-converting enzyme with angiotensin receptor blockers (ARBs) such as Sartans induces potent regulatory T cells and modulates TH1- and TH17-mediated potency [9]. This study is a guide for investigating ARBs and peptide mimetic drugs as possible regulators in the immunotherapy of autoimmune diseases [10,11]. ANG II is involved in diverse pathological situations in terms of tissue remodeling, cell proliferation, migration, tissue invasion, or angiogenesis [12,13]. Thus, ANG II has also relevant in the metastasis of cancers, since a relationship between hypertension and cancer exists [14]. 

ANG II, the key pressor short peptide, has been a very well-studied peptide by us, and diverse ANG II analogues have been synthesized towards the design and synthesis of non-peptide mimetics as angiotensin II receptor blockers [15,16,17]. In particular, extensive structure activity, nuclear magnetic resonance, and fluorescence studies revealed the importance of the N-terminal domain and of the proline residue to stabilize the active conformation of angiotensin II [18,19,20,21]. In this study, we investigated the mechanism of action of peptide and nonpeptide ARBs at the molecular and pharmacological level, with the aim of understanding the details of their AT1 receptor interactions. Since ARBs (sartans) are, in essence, ‘ANG II look-alikes’, it is anticipated that they will bind to Neprilysin and the ACE2 enzyme, which is the receptor used by the COVID-19 virion for cell entry. In this regard, we have developed a new class of ARB called bisartans (Bis), which contain two biphenyltetrazole groups, enabling bivalent interactions with critical receptor-based positively charged groups, such as (1) R167 of the angiotensin AT1 receptor and (2) Zinc^2+^ at the active site of ACE2 and Neprilysin. Accordingly, bisartans could have a dual role against COVID-19 by (1) preventing ANG II-induced toxic effects (pulmonary edema/inflammation/cytokine storm) and (2) blocking ACE2-spike protein interactions, particularly involving R and Zn active sites, thereby inhibiting cell entry of the virion.

The beneficial role of Neprilysin (NEP) in Renin Angiotensin System is depicted in Figure 1. In particular, Angiotensinogen (1–255) is cleaved by renin in the circulation to generate inactive decapeptide angiotensin1 (Ang1). Angiotensin 1 is cleaved by ACE to yield the octapeptide angiotensin II and by Neprilysin enzyme to yield heptapeptide angiotensin (1–7). In contrast to ACE producing vasoconstrictor angiotensin II, Neprilysin(NEP), also a zinc depended protease, is an activator of the alternative RAS in the murine and kidney, producing beneficial vasodilator angiotensin (1–7) from Ang (1–10) and from Ang (1–9), which could lead to therapeutic strategies. Additionally, heptapeptides, angiotensin (1–7), and angiotensin (Ala1–7) (Alamandine) are produced by ACE2, which degrades inflammatory angiotensin II in the renin angiotensin system, counterbalancing the AT1 receptor/angiotensin II axis. The heptapeptides lack phenylalanine, which is the structural requirement in angiotensin II for vasoconstriction and hypertension, and the roles of ACE2 and Neprilysin in the RAS are beneficial [22,23]. This study investigates in silico the docking of sartans and bisartans with neprilysin, in an effort to understand the molecular mechanisms of these interactions.

## 2. Methods

### 2.1. Isolated Tissues

Rat isolated smooth muscle tissue bioassays were conducted, as described previously [24,25,26]. Dose-response data were plotted in the format dose/response vs. dose to identify linearity and then as Hill plots, to obtain levels of cooperativity from the slope (Hill coefficient). These protocols for analyzing and interpreting cooperativity in dose-response curves have been described in detail previously [25,26,27].

### 2.2. In Silico Studies

Docking of ligands to the neprilysin (NEP) (PDB ID: 5JMY) [28] was also performed using AutoDock Vina [29] with default parameters. Point charges and dihedral barriers were initially assigned according to the YAMBER14 force field [30]. However, YAMBER point charges were damped to mimic less polar Gasteiger charges used to optimize the AutoDock scoring function. Docking was performed using non-periodic (walled) boundaries that effectively confined ligands to an approximately 17 × 17 × 32 Å cuboid volume. The setup was implemented using the YASARA molecular-modeling program [31]. The best hits and ligand conformational poses, expressed as kcal/mol free energy of binding, resulting from a minimum of 100 runs per ligand, were reported. 

MD simulations were run with YASARA [31]. The setup included an optimization of the hydrogen bonding network to increase the solute stability, and a pKa prediction to fine-tune the protonation states of protein residues at the chosen pH of 7.4. NaCl ions were added with a physiological concentration of 0.9%, with an excess of either Na or Cl to neutralize the cell. After steepest descent and simulated annealing minimizations to remove clashes, the simulation was run for up to 120 ns using the AMBER 14 force field [32] for the solute, GAFF2 [33] and AM1BCC [34] for ligands, and TIP3P for water. The cutoff was 8 Å for van der Waals forces (the default used by AMBER) [35], and no cutoff was applied to electrostatic forces (using the Mesh Ewald algorithm) [36]. MD simulation of the docked Neprilysin-Bisartan complexes was carried out using an NPT ensemble with periodic boundaries at 311 K, including explicit transferable intermolecular potential with 3 points (TIP3P) solvation at liquid density (0.997 g/cc) using AMBER14 parameters. Sodium and chloride ions were added at physiological concentration (0.9 wt%), with an excess of either ion used to maintain system neutrality. The in-vacuo molecular dynamics relaxation of Bis-A occurred in the presence of ZnCl or CH3-guanidine. The MD simulations were run for 3.5 ns in vacuo at 298 K using AMBER-14 parameters. Binding energies and affinity constants should be viewed qualitatively and in relative terms—they highlight differences between ligands and do not take into account entropy differences.

## 3. Results and Discussion

### 3.1. Two-State Receptor and Biased Agonism

The octapeptide ANG II acts at AR1 receptors in smooth muscle and other tissues causing contraction and blood pressure elevation. The severe acute respiratory syndrome coronavirus 2 (SARS-CoV-2) infection involves a viral cell-entry mechanism via angiotensin-converting enzyme 2 (ACE2), implicating angiotensin as a target for potential therapies. Lowering ANG II levels with angiotensin-converting enzyme (ACE) inhibitors or with AR1 receptor blockers (ARBs) provides effective treatment of hypertension. ARBs demonstrate blocking effects at AR1 receptors, which are similar to the effects seen with certain synthetic analogs of ANG II in which the C-terminal Phe is replaced by an aliphatic amino acid such as Ile.

Initially, [Sar1Ile8]ANG II (sarilesin) was designated as a Type 1 desensitizing antagonist [different from competitive surmountable Type II antagonists such as [Sar1Tyr(Me)4]ANG II (sarmesin)], because its effects were insurmountable and reminiscent of the desensitization/tachyphylaxis brought on by supramaximal doses of agonist [24]. However, detailed investigation of dose-response data, transformed into Hanes–Woolf plots and Hill plots [25], has shown that sarilesin induces negative cooperativity (Hill coefficient nH < 1) signifying negative efficacy, which is synonymous with inverse agonism (Table 1). Activities of ANG II analogues are classified as follows: Sar-Arg-Val-Tyr-Ile-His-Pro-Phe (superagonist), Sar-Arg-Val-Tyr(Me)-Ile-His-Pro-Phe (Sarmesin, surmountable antagonist), and Sar-Arg-Val-Tyr-Ile-His-Pro-Ile (Sarilesin, insurmountable inverse agonist) (Table 2).

Moreover, whereas submaximal doses of ANG II result in positive cooperativity (nH > 1 < 2) invoking receptor dimerization with a consequent increase in affinity for ANG II [26], supramaximal doses of ANG II can invoke negative cooperativity and tachyphylaxis (Figure 1). Hill coefficients also show differences in tissue responsiveness to ANG II, with receptor avidity in the decreasing order: aorta, portal vein, and uterus (Table 2).

The prolonged effect of [Sar1Ile8]ANG II does not appear to be due to a slow dissociation rate of this ligand from the receptor, because binding-displacement curves for 125I-ANG II and 125I-[Sar1Ile8]ANG II are superimposable [26]. It seems likely, therefore, that the receptor is locked down as a result of slow dissociation of the second messenger, which is coupled to the ‘inverted’ state of the receptor (Figure 1).

It is worth noting that the receptor mechanisms shown are not exclusive to angiotensin. Receptor dimerization and negative efficacy (inverse agonism) considerations apply to all G proteins coupled with seven transmembrane domain receptors, including those for nonpeptide ligands [27]. Indeed, the threshold effects and tachyphylaxis effects commonly observed in many pharmacological assays are symptomatic of positive and negative cooperativity, respectively.

#### 3.1.1. Angiotensin Peptides and Antipeptides—Regulators of RAS

The replacement of the C-terminal Phe residue of ANG II with an aliphatic amino acid (Ile, Leu, Val, Ala) converts the molecule to an analog with inverse agonist properties. It was of interest to find out if such a molecule is a naturally occurring product encoded by the human genome and designed to counteract ANG II. The sequence of human mRNA, which is complementary to that for ANG II [37], was found to encode the ANG II analogue EGVTVHPV [and likewise the ANG III analogue GVTVHPV]. The C-terminal Val residue confers inverse agonist properties to these ‘antipeptides’, whereas the absence of the Arg side chain has only modest effects on activity. Accordingly, angiotensin ‘antipeptides’ may be produced as a natural countermeasure to angiotensin, by binding to the inverse agonist site (Figure 1).

ANG II has a fundamentally important role in the regulation of cardiovascular function, and, perhaps not surprisingly, its actions are controlled at multiple levels. Axis control points include genetic expression and enzymatic biosynthesis of components of the RAS, giving rise to ANG II, metabolism by ACE2 to the vasodilator peptide ANG II (1–7) acting at AT2 receptors, and signaling effects at AT1 receptors. The present findings add yet another potential level of control for physiological regulation of the RAS, via the production of endogenous antipeptides acting directly as inverse agonists at AT1 receptors. Angiotensin receptor blockers (ARB sartans) used to treat hypertension (see below) apparently bind to a site on the angiotensin receptor which evolved to accommodate naturally occurring peptide inverse agonists.

#### 3.1.2. Charge-Relay Systems

Evidence has been accumulated from structure-reactivity, NMR, and fluorescence lifetime studies which suggest that activation of AT1 receptors is driven by the formation of a charge-relay system (CRS) in ANG II, encompassing triad TyrOH-His-Phe carboxylate, which generates tyrosinate anion species for activating the receptor (Figure 2A). ANG II CRS mechanism is analogous to that in Serine proteases [38]. Notably, methylation of the Tyr hydroxyl group of the superagonist [Sar1]ANG II, or of the inverse agonist [Sar1Ile8]ANG II, leads in both cases to reversible surmountable antagonists, implying the Tyr hydroxyl has a critical role at both the agonist and the inverse agonist states of the receptor. It is the amino acid at position 8 that ultimately determines which receptor state is preferred for binding [39].

The N-terminal portion of ANG II is thought to have a largely supportive role in maintaining the CRS [40]. Accordingly, the role of Arg2 in ANG II is to help stabilize the CRS and mainly act as a chaperone of the C-terminal business end of the molecule [40,41]. Its role is essentially replaced or supplanted by R167 of the receptor when ANG II binds to its receptor (see below). The role of Asp at position 1 of ANG II is considered non-obligatory, and possibly detrimental, since its replacement in [Sar1]ANG II results in a superagonist. Crystallography studies [42,43] have implicated a CRS-like interaction for the binding of the ARB inverse agonist olmesartan to AR1 receptors, wherein an intermolecular CRS is formed with the Tyr35 residue of the receptor (Figure 2B). For ANG II, CRS formation involves an intramolecular interaction (Figure 2A). In another study by Asada et al. [44] the X-ray structure of ANG II bound to AT2 R revealed an interaction of the peptide with Met128, suggested to be a key for receptor activation. The crystal pose of this structure has shown furthermore a hydrogen bond between Tyr OH and PheCOO^−^ within the peptide, indicating the formation of tyrosinate species in line with our ANG II model, where tyrosine hydroxylate is interacting with the receptor [45].

#### 3.1.3. Angiotensin Receptor Blockers (ARB Sartans)

The first nonpeptide ARB to be discovered was the surmountable antagonist losartan, which is metabolized to the inverse agonist losartan carboxylate in the bloodstream. This transformation involves the conversion of an imidazole-based neutral hydroxymethyl group of losartan to the carboxylate found in losartan carboxylate (Figure 2A). Similar considerations apply to the neutral carboxamide and carboxylate versions of olmesartan, which are likewise a surmountable antagonist and an inverse agonist, respectively [42]. As outlined above, the peptide equivalents are the TyrOMe and TyrO^−^ species in surmountable sarmesin and insurmountable sarilesin, respectively. In other words, it is the carboxylate group in ARBs that defines the inverse agonist activity (Figure 2A), and it is the tyrosinate species in [Sar1Ile8]ANG II that has the same functional role [40,41]. However, the tyrosinate anion in ANG II has two functions: stimulating contraction (positive cooperativity) at submaximal doses and inducing inverse agonism (negative cooperativity) at supramaximal doses. The charge-relay system in ANG II creates two negative charges, Tyr O^−^ and COO^−^, the spacing of which is exactly mimicked by the two acid groups in ARBs (Figure 3). These anion pairs form salt bridges with Arg167 and Lys199 on the receptor [42,43,46] (Figure 3). Salt bridges can provide energy for receptor binding affinity, to both the agonist and inverse agonist binding states of the receptor (Figure 1). For the agonist state, the dimerization process, which results in amplification of the contractile response, requires the presence of the Phe ring of ANG II [which is notably absent in ARBs]. The triggering process possibly involves a quadrupolar interaction of the Phe ring of ANG II with the Trp84 ring on the receptor.

#### 3.1.4. SARS-CoV-2, ARBs, and Bisartans

SARS-CoV-2 (COVID-19) patients often present compromised lung function due to severe pulmonary edema. We have noted when conducting rat pressor assays that bolus injections of EC50 doses of ANG II, carried out at 10–20-min intervals, result after several hours in chronic pulmonary edema, as evidenced by the appearance of fluid backing up the tracheal tube. Pulmonary edema seen in COVID-19 infection should respond to ARBs counteracting the effect of toxic ANG II. In addition, ARBs, which are angiotensin look-alikes, should bind to ACE2 and may disrupt the binding of the COVID-19 spike protein, which is vital for the cell entry of the virus [15,47,48]. A recent study reported the discovery and facile synthesis of a new class of sartan-like arterial antihypertensive drugs (angiotensin receptors blockers [ARBS], referred to as ‘bisartans’ [16]. Bisartans are novel bis-alkylated imidazole sartan derivatives bearing dual symmetric inionic biphenyl tetrazole moieties, which in in silico studies exhibited higher binding affinities for the ACE2/spike protein complex (PDB ID: 6LZG), compared to all other known sartans. Bisartans are classified as BisA, B, C, and D, where BisA and B bear the butyl group of position 2 in the imidazole ring, and BisC and D are at position 4, as in losartan [16]. In another study, we further investigated the structural hydropathy traits and polarity properties of mutants that induce the SARS-CoV-2 dominant mutation, which increase transmissibility and infectivity [16]. Arginine blockers, such as ARBs and bisartans, which bear anionic tetrazoles and carboxylates, were found in in silico studies to be ideal candidate drugs for viral infection by weakening S-protein RBD binding to ACE2 and discouraging hydrolysis of cleavage sites [16]. An example of the strong binding of BisA with methylguanidine is where amino groups of guanidine moiety crab the two tetrazoles of BisA. In a similar manner, the two tetrazoles of BisD are crabbed by Zn^+2^ (Figure 4 and Figure 5).

#### 3.1.5. Bisartans Bound to AT1R

We have developed a new generation of ARBs, called bisartans [15,49]], the structures of which include two tetrazole groups (Figure 6). Bisartans A and B bear the butyl group at position 4 of the imidazole ring, while bisartans C and D are at position 2 as in losartan. Bisartans are potent insurmountable antagonists at AR1 receptors and contain powerful chelators of the zinc atom at the active site of zinc proteases such as ACE2. Coordination of ACE2 Zn^+2^ with ANG II and analogues involving four residues D, Y, H, and the C-terminal carboxylate has been reported [50]. Knowing that tyrosinate in sarilesin and carboxylate in EXP3174 were both effective at salt bridging to R167, we wondered if tetrazolate, which is a functional mimetic of carboxylate, might also suffice. By using imidazole as the template for mounting two biphenyl tetrazole (BPT) groups, we conjectured that the resulting positive charge on the imidazole should be at about the right distance from the two tetrazoles to subsume the space normally occupied by the Arg2 guanidino group of ANG II, when the peptide is bound to the receptor (Arg2 is predicted to interact with D263 and D281 of the receptor [51]). It turned out that bisartans are indeed potent insurmountable antagonists at AT1 receptors [52] and are the first ARBs discovered that do not use a carboxylate for binding to Arg167 but instead use a second tetrazolate to form the critical salt bridge with Arg167, which appears to provide for insurmountable activity. Bisartans likely exhibit significant bioactivity at AT1 receptors because of tight binding to R167 due to the particular geometry of the double tetrazole unit surrounding the R167 guanidino group. Essentially, bisartans can enfold the R167 guanidino group in the tight embrace of two tetrazole moieties [52]. Interactions between receptor and ARBs were determined from the crystal structure reported by Zhang et al. [42,43]. The critical interactions of olmesartan are with R167, W84, Y35, and K199 residues of AT1R (Figure 6).

#### 3.1.6. Bisartans Bound to NEP

Here we report new in silico results suggesting bisartans also undergo specific binding to the zinc-dependent endopeptidase NEP, which plays a pivotal role in regulating humoral levels of beneficial vasoactive polypeptides, including atrial and brain natriuretic peptides, bradykinin, adrenomedullin, and endothelin-15 [53,54]. Inhibition of NEP causes increased levels of the natriuretic peptides and, thus, represents an important therapeutic target for controlling blood pressure and heart disease in humans. In the present study, a series of 17 conventional angiotensin receptor blockers (ARBs), including four novel bisartans (BisA–BisD) and two potent NEP antagonists, LBQ657 and Candoxatrilat, were docked to the soluble extracellular domain of NEP (PDB ID: 5JMY). Results indicated that AutoDock Vina accurately reproduced the correct X-ray crystallographic posture of LBQ657 in the main Zn^2+^ pocket of NEP with a root-mean-square deviation (RMSD) ~0.01Å. Dissociation constants (Kd) for BisA and LBQ657 (32,095.31 and 49,026.0703, respectively) were in agreement with their binding energies, although the per-atom binding efficiency for LBQ657 0.3561 kcal/(mol*Atom) was somewhat greater than that of BisA 0.3561 kcal/(mol*Atom).

Of the 17 ligands evaluated, the bisartan BisA exhibited the strongest binding, with two other bisartans (C, D) ranking in the top five scoring positions (Figure 7B). BisA binding was stabilized by hydrophobic and π–π resonance interactions of the drug with contacting NEP residues, as well as cation–π interactions between a terminal tetrazole group and the zinc ion. Molecular dynamics simulations carried out for approximately 40 ns at 311 K in physiological saline demonstrated the solvated NEP-BisA complex (PDB 5JMY-BisA) was stable (average drug RMSD = 1.20 A) over this period (Figure 8). A nearly identical result (average BisA RMSD = 1.64 Å) was obtained for a longer 90 ns MD simulation run under the same conditions, but using a complex comprised of BisA docked to the zinc pocket of human neprilysin (PDB 6GID) (data not shown).

The initial docked conformation of BisA (at t = 0 ns) underwent a conformational change during the initial 300 ps of an MD simulation, causing one tetrazole group of BisA to approach the zinc ion resulting in a stabilizing ionic (cation–pi) bonding interaction. This tetrazole group also formed additional ionic and hydrogen-bonding interactions with Arg717 (Figure 9).

#### 3.1.7. ARBs: Potential Antiviral Drugs for the Treatment of COVID-19

This study was based on the knowledge that cardiovascular disease is entirely related to COVID-19, in terms of mechanisms that trigger the disease [55]. The storm of cytokines released in COVID-19 patients with pneumonia is related to the overexpression of toxic ANG II in the renin–angiotensin system (RAS) [56,57], and clinical studies have shown that morbidity and mortality rate was lower in hypertensive patients infected by SARS-CoV-2 who are taking angiotensin-converting enzyme inhibitors and ANG II-receptor blockers (ARBs), compared to patients not taking these drugs [58,59,60,61,62,63,64,65,66,67]. ANG II, the major RAS component, and SARS-CoV-2 spike protein cleavage, by furin and 3CL, which initiate infection, operate through similar charge-relay-system mechanisms. Tyrosinate in ANG II, serinate in furin, and cysteinate in 3C-like protease (3CLpro) are anions created through the charge-relay-system mechanism and trigger activity via their nucleophile anions. Similar charge-relay-system mechanisms spark and mediate the action of ANG II [45] and the serine-like proteases furin and 3CL [68,69,70,71]. Bisartans were found to bind strongly to the SARS-CoV-2 RBD/ACE2 complex with higher affinity compared to other common sartans. In a similar mode as in the interaction of ARBs with AT1R, bisartans block the critical aminoacid arginine in the furin cleavage site 681–686, which catalyzes the cleavage of the spike protein and triggers COVID-19. Mutation P681R, as in the Delta variant, increases the basicity of the rich arginine cleavage site and the severity of the infection [47]. Bisartans have also been found to bind in silico to 3CL protease such as Pfizer’s Paxlovid combination drugs (PF-07321332 and Ritonavir) [68,69,70]. However, in the COVID-19 assay, BisD was found to be less effective [16,72]. The catalytic center of furin is the triad Asp–His–Ser, and for the 3CL protease, it is the dyad Cys145–His41 [65]. Bisartans act at the three cell entries (ACE2, Furin, 3CLpro) and are ideal potential antiviral drugs for the treatment of COVID-19.

#### 3.1.8. Effects of ACE, ACE2, and NEP in the Renin–Angiotensin System

The above findings with NEP may be extrapolated to the zinc metalloprotease ACE2, which is the receptor used by the COVID-19 virion to enter host cells. This is justified, as both NEP and ACE2 are zinc proteases. Schematic representation of the renin–angiotensin system shows the same beneficial effect of the two proteases, as both produce vasodilatory Ang(1–7) [73] (Figure 1). Accordingly, it is anticipated that bisartan could bind to the zinc atom in ACE2 and may allosterically inhibit binding and cell entry of the virion (which binds to a different site on the enzyme). In COVID-19, ARBs are known to alleviate some of the symptoms of infection. Bisartans could have a dual role by (1) blocking ANG II induced toxic effects (pulmonary edema/inflammation/cytokine storm) and (2) preventing ACE2–spike protein interactions, particularly involving R and Zn active sites, thereby inhibiting cell entry of the virion.

## 4. Conclusions

Two states of the AT1 receptor, agonist and inverse agonist states, interact, respectively, with surmountable and insurmountable ARBs and angiotensin analogues. The inverse agonist state can bind a naturally occurring angiotensin ‘antipeptide’, which is encoded by mRNA complementary to that for human ANG II. Agonist activation of the receptor may involve CRS exchange of the Tyr4 of ANG II with Y35 of the receptor [52], whereas inverse agonism is characterized by salt-bridge formation between the ligand and R167 of the receptor [52]. Accordingly, ARB sartans interact preferentially with R167 cation of the AT1R, resulting in the switching of the receptor into its inverted agonist state, and bisartans have the opportunity for bivalent (2 tetrazoles) bonding and, thereby, a stronger interaction leading to increased antihypertensive activity.

Tetrazole can also bind/chelate divalent cations found at the active site of metalloproteases, and Recife model 1 (RM1) energy calculations for the bivalent interaction (both tetrazoles) of bisartan with Zn^2+^ (−165 kcal/mol) or with the guanidine group of Arg (−106 kcal/mol) are relative to a C-C covalent bond (−330 kcal/mol). In silico studies reveal that bisartans can chelate the active site zinc atom of the enzyme NEP, although only one of the two tetrazoles was found to bind to zinc (Figure 9B), either due to steric hindrance preventing accessibility of the second tetrazole, to compensating multiple secondary interactions, or to limitations of the modeling software/computing power. Visual inspection of the 3D structure in Figure 9 suggests that there is likely enough space to accommodate two tetrazole groups, which is perhaps not surprising since the zinc pocket is designed to bind elongated peptide sequences of natural substrates. Tetrazole is a weak acid [and a functional mimetic of carboxyl, pKa~4], which will bind to protein basic groups in the order Zn^2+^ > Arg > Lys > others, so there is considerable energy to be gained from the binding of both tetrazoles to zinc. On the other hand, the positive charge on the central imidazole ring of bisartan also has to be taken into account and is shown here (Figure 9B) bonded to F544 of NEP in a parallel-plate interaction (as would be expected for a positively charged imidazole ring bonding to the Phe ring quadrupole). In the initial docking (Figure 9A), R102 and R110 outcompete the zinc ion, but, after relaxing the system (via MD simulation), this tetrazole reorients to interact with zinc (Figure 9B). Why only one tetrazole interacts with zinc is probably best explained by considering that once one tetrazole is captured by zinc, the other tetrazole has to climb out of an energy well to wrap back around the zinc ion. This may be energetically difficult since the butyl chain and biphenyl of bisartan are stabilized by hydrophobic interactions with I535, Y545, G548, T206, F544, and H587 (Figure 9A). Collectively, these interactions, together with the central imidazole, discourage or preclude the dihedral rotations needed to move the second tetrazole into a position where it might begin to interact with zinc.

## Data Availability

Not applicable.

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
