# Peer review of "Actions of Novel Angiotensin Receptor Blocking Drugs, Bisartans, Relevant for COVID-19 Therapy: Biased Agonism at Angiotensin Receptors and the Beneficial Effects of Neprilysin in the Renin Angiotensin System"

_molecules, 2022, doi:10.3390/molecules27154854_

Round 1
Reviewer 1 Report
Sir,
readability of the paper should be greatly improved, starting - for instance - from the first three lines of Introduction: ‘Angiotensin is a peptide hormone that mediates the main actions of…’ Write Angiotensin II, 'angiotensin', as such, means nothing. Delete the following sentence that mentions hypertension and angionin, etc.
Rushing to the end of the paper, the authors' extrapolation of the findings with NEP to zinc metallopeptidase ACE2 seems to me wishful thinking. I mean, the real crux of the matter, when trying to impede SARS-CoV-2 binding to the extracellular domain of ACE2, consists in the preservation of ACE2 ability to generate Ang(1-7) out of Ang II (whatever the pharmacological manoeuvre employed to stop viral binding to ACE2). In fact, it has been known for years how to impede coronavirus sticking to cell membrane ACE2 (see Fowler et al, 2004). Far more important is the preservation of ACE2 function while coronavirus binding to this zinc metallopeptidase is somehow prevented.
Between the above-mentioned Introduction and these Conclusions, the paper is still very hard to read.
Author Response
We are grateful to the reviewers 1&2 for their valuable comments. We fully agree with all their comments, and we improved the current manuscript according to their instructions. We hereby present comments of the reviewers and our responses to them as well the revised article.
Reviewer 1
Comment
readability of the paper should be greatly improved, starting - for instance - from the first three lines of Introduction: ‘Angiotensin is a peptide hormone that mediates the main actions of…’ Write Angiotensin II, 'angiotensin', as such, means nothing. Delete the following sentence that mentions hypertension and angionin, etc.
Reply
As suggested Angiotensin is replaced Angiotensin II.
Also sentence that mentions hypertension and angionin is deleted.
Comment
Rushing to the end of the paper, the authors' extrapolation of the findings with NEP to zinc metallopeptidase ACE2 seems to me wishful thinking. I mean, the real crux of the matter, when trying to impede SARS-CoV-2 binding to the extracellular domain of ACE2, consists in the preservation of ACE2 ability to generate Ang(1-7) out of Ang II (whatever the pharmacological manoeuvre employed to stop viral binding to ACE2). In fact, it has been known for years how to impede coronavirus sticking to cell membrane ACE2 (see Fowler et al, 2004). Far more important is the preservation of ACE2 function while coronavirus binding to this zinc metallopeptidase is somehow prevented.
Between the above-mentioned Introduction and these Conclusions, the paper is still very hard to read.
Reply
- On the ACE2 function comment
We cannot find any paper on ACE2 by Fowler et al for the year 2004.
“It is not known if preventing the virus from binding to ACE2 will interfere significantly with the normal physiological function of ACE2. What we know from clinical studies is that Sartans protect hypertensive patients infected by Covid by preventing the virus from binding to ACE2, this shown by docking and antiviral studies (16). Yet Sartans allow ACE2 to exert its normal physiological antihypertensive vasodilatory functions which is to degrade angiotensin II and produce Ang(1-7)”
- On the extrapolation of NEP findings to ACE2 comment
In the second line of last paragraph we inserted after “host cells” the following sentence.
“This is justified as both NEP and ACE2 are zinc proteases. Schematic representation of the renin angiotensin system shows the same beneficial effect of the two proteases as both produce vasodilatory Ang(1-7)”( Scheme 1 to be included). Accordingly………
- On the comment that paper is hard to read, two paragraphs were added at the end of the Introduction and the Results and Discussion section, explaining the reasoning of NEP /ARBs docking. Also a scheme 1 to demonstrate the beneficial effects of Neprilysin.
The beneficial role of Neprilysin (NEP) in Renin Angiotensin System is depicted in Scheme1. In particular Angiotensinogen(1-255) is cleaved by renin in the circulation to generate inactive decapeptide angiotensin1(Ang1). Angiotensin 1 is cleaved by ACE to yield the octapeptide angiotensin II and by Neprilysin enzyme to yield heptapeptide angiotensin(1-7). In contrast to ACE producing vasoconstrictor angiotensin II, Neprilysin(NEP), also a zinc depended protease, is an activator of the alternative RAS in the murine and kidney, producing beneficial vasodilator angiotensin (1-7) from Ang(1-10) and from Ang(1-9) which could lead to therapeutic strategies. Additionally heptapeptides, angiotensin(1-7) and angiotensin (Ala1-7) (Alamandine) are produced by ACE2 which degrades inflammatory angiotensin II in the renin angiotensin system counterbalancing the AT1 receptor/ angiotensin II axis. The heptapeptides lack phenylalanine which is the structural requirement in angiotensin II for vasoconstriction and hypertension and the roles of ACE2 and Neprilysin in the RAS are beneficial [22,23]. This study investigates in silico the docking of sartans and bisartans with neprilysin in an effort to understand the molecular mechanisms of these interactions.
In addition the revised manuscript is read by the native English speaking and we believe that meets all the linguistic requirements.
Reviewer 2 Report
In their manuscript entitled, “Actions of Novel Angiotensin Receptor Blocking Drugs, Bisartans, Relevant for COVID-19 Therapy: Biased Agonism at Angiotensin Receptors’’ Moore et al. reported an in-silico investigation of the binding of a class of angiotensin receptor blockers called Bisartans to the active site zinc atom of the endopeptidase Neprilysin. Here, in-vivo and modeling results suggest Bisartans can inhibit ANG II-induced pulmonary edema. The following concerns should be addressed prior to the publication of the paper.
Line 137-138: Please fix the usage of “(” to avoid confusion. Please clarify the number of runs, what temperature or pressure is selected, and what ensemble (NPT, NVT, ..) and thermostat method have been used for the MD simulations.
Figure 4: Please clarify the in-vacuo molecular dynamics relaxation method description in sec. 2 (Methods) in more detail. Ideally, any method description should be placed in a designated method section.
Line 408: Please clarify the MD simulation total run-time of 40 ns that may appear inconsistence with line 135 stating the total run-time of 120 ns. If the MD simulation run-time is 120 ns, please further verify if the ligand stays bound in the entire of its corresponding MD simulation and did not drift away into the solvent.
Line 417: Please fix the PDB ID “5JM”, is it “5JMY”?
Line 419: Please consider presenting a figure showing the correct prediction of the x-ray pose of the bound inhibitor LBQ657 in a supplementary information document.
Figure 7: Please clarify the actual values of dissociation constants for BisA and LBQ657, it seems they are relatively close that may appear inconsistence with lines 403-404 that BisA exhibits the strongest binding. Ideally, all compounds' dissociation constants values and 2D interaction ligand-receptor diagrams of docking poses should be presented at least in a supplementary information document.
Author Response
We are grateful to the reviewers 1&2 for their valuable comments. We fully agree with all their comments, and we improved the current manuscript according to their instructions. We hereby present comments of the reviewers and our responses to them as well the revised article.
Reviewer 2
Comment
In their manuscript entitled, “Actions of Novel Angiotensin Receptor Blocking Drugs, Bisartans, Relevant for COVID-19 Therapy: Biased Agonism at Angiotensin Receptors’’ Moore et al. reported an in-silico investigation of the binding of a class of angiotensin receptor blockers called Bisartans to the active site zinc atom of the endopeptidase Neprilysin. Here, in-vivo and modeling results suggest Bisartans can inhibit ANG II-induced pulmonary edema. The following concerns should be addressed prior to the publication of the paper.
Line 137-138: Please fix the usage of “(” to avoid confusion. Please clarify the number of runs, what temperature or pressure is selected, and what ensemble (NPT, NVT, ..) and thermostat method have been used for the MD simulations.
Reply
The usage of “(”was fixed in the text. The additional infos were added in the text.
Comment
Figure 4: Please clarify the in-vacuo molecular dynamics relaxation method description in sec. 2 (Methods) in more detail. Ideally, any method description should be placed in a designated method section.
Reply
Now a more detailed description of MDs was added in method section.
Comment
Line 408: Please clarify the MD simulation total run-time of 40 ns that may appear inconsistence with line 135 stating the total run-time of 120 ns. If the MD simulation run-time is 120 ns, please further verify if the ligand stays bound in the entire of its corresponding MD simulation and did not drift away into the solvent.
Reply
The simulation was run for up to 120 ns. In the case of the docked Neprilysin-BisA complex, the average RMSD for the BisA drug was 1.20 Å from 1 ns to 39.7 ns, suggesting it was stably bound in the zinc pocket for the duration of the 40-ns.
Comment
Line 417: Please fix the PDB ID “5JM”, is it “5JMY”?
Reply
Now, it is corrected in the text.
Comment
Line 419: Please consider presenting a figure showing the correct prediction of the x-ray pose of the bound inhibitor LBQ657 in a supplementary information document.
Reply
A revised Figure 9, which now includes the superimposed postures of the experimental x-ray pose of the known NEP inhibitor LBQ657 compared to the drug pose predicted (calculated) by AutoDock VINA.
Comment
Figure 7: Please clarify the actual values of dissociation constants for BisA and LBQ657, it seems they are relatively close that may appear inconsistence with lines 403-404 that BisA exhibits the strongest binding. Ideally, all compounds' dissociation constants values and 2D interaction ligand-receptor diagrams of docking poses should be presented at least in a supplementary information document.
Reply
Addition of a new sentence at lines 410-412 to address the comment about confirming the binding energies of BisA with LBQ657.
Round 2
Reviewer 2 Report
While the implemented changes are much appreciated, the following concern should be addressed prior to the publication of the paper.
Lines 434-435: Please clarify the state of docked Neprilysin-BisA complex during MD simulation from 40 ns to 120 ns. Did the ligand stay bound from 40 ns to 120 ns? If the ligand did drift away into the solvent, please comment on any potential reason or limitation.
Figure 8: The figure shows only 40 ns of the MD simulation, and the data for 40 ns to 120 ns is missing. Please consider showing the data for the entire 120 ns MD simulation, including snapshots and RMSD plot for the entire 120 ns. Additionally, please consider adding the unit to the numbers above each snapshot (7.9 ns instead of 7.9).
Author Response
Reviewer 1
Sir,
readability of the paper should be greatly improved, starting - for instance - from the first three lines of Introduction: ‘Angiotensin is a peptide hormone that mediates the main actions of…’ Write Angiotensin II, 'angiotensin', as such, means nothing. Delete the following sentence that mentions hypertension and angionin, etc.
Rushing to the end of the paper, the authors' extrapolation of the findings with NEP to zinc metallopeptidase ACE2 seems to me wishful thinking. I mean, the real crux of the matter, when trying to impede SARS-CoV-2 binding to the extracellular domain of ACE2, consists in the preservation of ACE2 ability to generate Ang(1-7) out of Ang II (whatever the pharmacological manoeuvre employed to stop viral binding to ACE2). In fact, it has been known for years how to impede coronavirus sticking to cell membrane ACE2 (see Fowler et al, 2004). Far more important is the preservation of ACE2 function while coronavirus binding to this zinc metallopeptidase is somehow prevented.
Between the above-mentioned Introduction and these Conclusions, the paper is still very hard to read.
Reviewer 2
In their manuscript entitled, “Actions of Novel Angiotensin Receptor Blocking Drugs, Bisartans, Relevant for COVID-19 Therapy: Biased Agonism at Angiotensin Receptors’’ Moore et al. reported an in-silico investigation of the binding of a class of angiotensin receptor blockers called Bisartans to the active site zinc atom of the endopeptidase Neprilysin. Here, in-vivo and modeling results suggest Bisartans can inhibit ANG II-induced pulmonary edema. The following concerns should be addressed prior to the publication of the paper.
Line 137-138: Please fix the usage of “(” to avoid confusion. Please clarify the number of runs, what temperature or pressure is selected, and what ensemble (NPT, NVT, ..) and thermostat method have been used for the MD simulations.
The usage of “(”was fixed in the text. The additional infos were added in the text.
Figure 4: Please clarify the in-vacuo molecular dynamics relaxation method description in sec. 2 (Methods) in more detail. Ideally, any method description should be placed in a designated method section.
Now a more detailed description of MDs was added in method section.
Line 408: Please clarify the MD simulation total run-time of 40 ns that may appear inconsistence with line 135 stating the total run-time of 120 ns. If the MD simulation run-time is 120 ns, please further verify if the ligand stays bound in the entire of its corresponding MD simulation and did not drift away into the solvent.
In this study, MD simulations were carried out for periods ranging from about 40ns to 120ns, depending on the system. In the case of the docked Neprilysin-BisA complex (i.e., PDB 5JMY-BisA), the MD simulation described was run for 39.7ns. The drug (BisA) was stable over the duration of this run with an average RMSD = 1.20 Angstroms. A nearly identical result (average BisA RMSD = 1.64 Å) was obtained for a longer 90-ns MD simulation run under the same conditions, but using a complex comprised of BisA docked to the zinc pocket of human neprilysin (PDB 6GID) (data not shown). These results are clarified in lines 418-423.
The simulation was run for up to 120 ns. In the case of the docked Neprilysin-BisA complex, the average RMSD for the BisA drug was 1.20 Å from 1 ns to 39.7 ns, suggesting it was stably bound in the zinc pocket for the duration of the 40-ns.
Line 417: Please fix the PDB ID “5JM”, is it “5JMY”?
Now, it is corrected in the text.
Line 419: Please consider presenting a figure showing the correct prediction of the x-ray pose of the bound inhibitor LBQ657 in a supplementary information document.
Additions by Harry – this is shown in Figure 9A
Figure 7: Please clarify the actual values of dissociation constants for BisA and LBQ657, it seems they are relatively close that may appear inconsistence with lines 403-404 that BisA exhibits the strongest binding. Ideally, all compounds' dissociation constants values and 2D interaction ligand-receptor diagrams of docking poses should be presented at least in a supplementary information document.
Additions by Harry – added text clarifying BisA and LBQ657 Kd values – see lines 410-412